# The macroecology of the COVID-19 pandemic in the Anthropocene

**Piotr Skórka**[1]*, **Beata Grzywacz**[2], **Dawid Moroń**[2], **Magdalena Lenda**[1]

**1** Institute of Nature Conservation, Polish Academy of Sciences, Kraków, Poland, **2** Institute of Systematics and Evolution of Animals, Polish Academy of Sciences, Kraków, Poland

* skorasp@gmail.com

**Data Availability Statement:** All relevant data are within the manuscript and its Supporting Information files.

**Funding:** The authors received no specific funding for this work

## Abstract

Severe acute respiratory syndrome coronavirus 2, the virus that causes coronavirus disease 2019 (COVID-19), has expanded rapidly throughout the world. Thus, it is important to understand how global factors linked with the functioning of the Anthropocene are responsible for the COVID-19 outbreak. We tested hypotheses that the number of COVID-19 cases, number of deaths and growth rate of recorded infections: (1) are positively associated with population density as well as (2) proportion of the human population living in urban areas as a proxies of interpersonal contact rate, (3) age of the population in a given country as an indication of that population's susceptibility to COVID-19; (4) net migration rate and (5) number of tourists as proxies of infection pressure, and negatively associated with (5) gross domestic product which is a proxy of health care quality. Data at the country level were compiled from publicly available databases and analysed with gradient boosting regression trees after controlling for confounding factors (e.g. geographic location). We found a positive association between the number of COVID-19 cases in a given country and gross domestic product, number of tourists, and geographic longitude. The number of deaths was positively associated with gross domestic product, number of tourists in a country, and geographic longitude. The effects of gross domestic product and number of tourists were non-linear, with clear thresholds above which the number of COVID-19 cases and deaths increased rapidly. The growth rate of COVID-19 cases was positively linked to the number of tourists and gross domestic product. The growth rate of COVID-19 cases was negatively associated with the mean age of the population and geographic longitude. Growth was slower in less urbanised countries. This study demonstrates that the characteristics of the human population and high mobility, but not population density, may help explain the global spread of the virus. In addition, geography, possibly via climate, may play a role in the pandemic. The unexpected positive and strong association between gross domestic product and number of cases, deaths, and growth rate suggests that COVID-19 may be a new civilisation disease affecting rich economies.

**Competing interests:** The authors have declared that no competing interests exist

## 1. Introduction

Macroecology is the study of broad-scale ecological patterns and processes [1]. Few ecologists, however, study the influence of the environment on humans, including the effects of biotic, abiotic, and social conditions on the population growth, economy, and health of our own species [2,3]. The emerging discipline of human macroecology [3] has an interesting duality [2]. The Homo sapiens is one of the most powerful species to inhabit the Earth [2] and is now a major geological and environmental force, as important as, or more important than, natural forces [4]. Thus, it has been suggested that the Earth is in the epoch called Anthropocene [4,5]. However, humans are subject to the same biological laws as any other organism. One of the most important areas of macroecology in the human context is disease ecology [6,7]. Humans, as hosts, exhibit three specific macroecological patterns: (1) humans spreading geographically disperse pathogens and parasites, (2) humans visiting or settling in new areas encounter new organisms, including new pathogens, and new alternative hosts for existing pathogens and parasites; (3) increased human population density and frequency of contact substantially influence the ecology of disease [2]. Thus, understanding how the spread of diseases is related to environmental and socioeconomic factors requires a global perspective [8].

New infectious diseases determine changes in mortality in populations of all organisms, including humans [9,10]. Among many viral diseases in humans, those caused by coronaviruses are especially troublesome [11]. Coronaviruses are a large family of viruses that usually cause disease in wild animals, but several of them, probably including severe acute respiratory syndrome coronavirus 2 (SARS-CoV-2), have made the jump to humans [11]. New viruses may be a threat to health systems and economies, and may even cause pandemics [12].

Coronavirus disease (COVID-19), caused by the SARS-CoV-2 virus, has been present since mid-December 2019. The first case of coronavirus was probably earlier (on 17 November, according to government data reported in the *South China Morning Post*), but until December, Chinese officials did not know that they had a new type of virus [13]. The World Health Organization officially recognised this disease on 11 March 2020 as a global pandemic [14]. In December and January, the incidence was limited primarily to the city of Wuhan in central China, but as early as mid-January, the virus quickly spread throughout China. On 13 January 2020, the first case outside China was confirmed. On 24 January, the first case was reported in Europe. In the second half of February, outbreaks with hundreds of patients erupted in South Korea, Italy, and Iran. On 20 June, the number of infected people worldwide reached over 8,385,440, of which 450,686 died [15]. Coronavirus-infected patients were registered on all continents, except Antarctica.

The COVID-19 pandemic will probably have numerous effects on the functioning of the human population, and, consequently, vast ecological consequences for human-affected ecosystems (e.g. bans to wildlife trade and increased poaching) [16]. It is thus urgent to recognise the factors responsible for the spread of this pathogen among human societies. This novel virus is unaffected by any immunity that people may have to older strains and can, therefore, spread extremely rapidly and infect very large numbers of humans in a short period of time. Typically, the SARS-CoV-2 virus is transmitted from infected individuals through the air by coughs or sneezes, creating aerosols containing the virus or by contact with contaminated surfaces, where the virus can survive for hours to days at a time [17]. Therefore, population density should positively correlate with the number of infections, deaths, and growth rate of infection cases. Higher population density increases the number of contacts among individuals and thus may mediate the transmission of pathogens [18,19]. The highest human population density occurs in urban areas. Towns and cities are also the usual areas of numerous social contact [20]. The high density of cars, buildings, and factories increases environmental

pollution in urban areas compared with rural ones. This imposes additional stress on the immune system [21]. Thus, it may be expected that pandemics are most common in urbanised countries.

Disease spread increases with the exchange of people between human populations. In the globalisation era, people increasingly change their location [21,22]. International travel has connected the world in the past century, and this mobility facilitates coronavirus transmission, allowing regional epidemics to become worldwide pandemics within a matter of weeks or even days. The mass movement of large numbers of people creates new opportunities for the spread and establishment of common or novel infectious diseases [23,24]. Thus, one may predict that a higher number of tourists and the net immigration rate should be positively associated with COVID-19 cases.

Models predict that children can transmit different types of viruses [25,26]. The higher frequency of disease incidence among children and young adults than that in the older population is mainly attributable to a low level of immunity in these age groups due to lower past exposure to infectious diseases [27]. However, studies on H1N1 swine flu cases during the late spring and summer of 2009 in various countries showed a substantial age shift in local transmission cases, with adults mainly responsible for seeding unaffected regions and children most frequently driving community outbreaks [28]. A low number of acute courses of COVID-19 cases in young people indicates that young people may be vectors of COVID-19 for additional transmission. Thus, it may be expected that the number of cases may be higher in countries with lower average life spans. On the other hand, older people have a weaker immune response and poorer general health, and are affected the most by COVID-19 and other viruses [29]. Thus, one may expect that the number of deaths will be the highest in countries with a high average life span.

In addition, other socioeconomic factors may be associated with the prevalence of pathogens. Marginal and disadvantaged people with low socioeconomic status are generally more vulnerable during a pandemic outbreak of disease [30]. Limited access to the media, lack of adequate resources for precautionary activities, lower literacy rates, inadequate access to health services, and crowded accommodations make people more prone to be affected by the pandemic [31]. Gross domestic product (GDP) is a commonly used indicator of socioeconomic variables [32]. For example, GDP correlates positively with the healthcare system and the probability of survival of people with dangerous diseases such as cancer [33]. Hence, it is expected that the number of cases, deaths, and rates of infection growth should be negatively associated with GDP.

In this paper, we aim to determine which global factors are associated with the early pandemic of COVID-19. We tested the hypotheses that the number of infections, deaths, and the rate of growth in the number of COVID-19 infections are:

1. Positively associated with human population density.

2. Positively associated with the proportion of the population living in urban areas.

3. Negatively associated with the median age of the human population. However, the number of deaths should be positively associated with the median age of the population.

4. Positively associated with the number of tourists visiting a given country

5. Positively associated with the net migration rate (proportion of immigrants) in a given country.

6. Negatively associated with gross domestic product.

We tested these hypotheses by including variables that are inevitably related to pandemic spread, such as number of days since the start of the pandemic in a given country, global locality (geographic coordinates of the centroid of each country).

## 2. Methods

### 2.1. Data

We used publicly available databases. Data on COVID-19 were downloaded from the European Centre for Disease Prevention and Control (https://www.ecdc.europa.eu/en/publications-data/download-todays-data-geographic-distribution-covid-19-cases-worldwide) on 12 April 2020.

Data of socioeconomic variables in each country where COVID-19 infections were reported were derived from the United Nations Population Division available via Worldmeters (https://www.worldometers.info/world-population/population-by-country/), downloaded on 18 March 2020.

Data on the number of tourists were obtained from IndexMundi (https://www.indexmundi.com/facts/indicators/ST.INT.ARVL/rankings).

Moreover, data on geographic coordinates of country centroids was downloaded on 18 March and 25 May 2020 from WorldMap (https://worldmap.harvard.edu/data/geonode:country_centroids_az8).

Data were compiled and analysed in R Environment [34] with the set of packages in 'tidyverse' [35]. All data and codes are available in the Supplementary material.

### 2.2. Data analysis

We analysed three response variables: 1) the number of COVID-19 cases, 2) the number of deaths due to the infection, and 3) growth rate of the infection cases. The growth rate of infection was determined by fitting the exponential growth curve for data in each country. The explanatory variables were: human population density (Dens), the proportion of the population living in urban areas (Urban), median age of the population (Age), number of tourists visiting a country (Tour), net migrations rate (Mig; negative value if emigration prevails, positive if immigration prevails), gross domestic product in millions of US dollars (GDP), time in days since the first case recorded in a given country (Time), geographic longitude (Lon), and latitude (Lat) of a country centroid.

Gradient boosting regression trees (GBRTs) [36] implemented in 'h2o' package version 3.30.0.1 [37] were used to analyse the relationships between the explanatory variables and dependent variables. Gradient boosting regression trees are efficient machine learning algorithms that have been proven successful across many domains and are among the leading machine learning algorithms [38–40]. Boosting improves model accuracy by searching for many rough prediction rules rather than the single most accurate prediction rule [39,40]. Gradient boosting regression trees generate a final model that is more robust than a single regression tree model and enables curvilinear functions to be modelled [41]. Another advantage of this method is that it copes with collinearity among variables [41], which was the case in our dataset (Fig 1). Gradient boosting regression trees calculate the relative importance of explanatory variables [40,41] in the predictive model rather than P-values, which have been criticised [42,43].

The GBRTs are prone to overfitting, but this can be solved by tuning the parameters [40]. The settings of the GBRTs model were tuned by searching for the optimal set of parameters minimising the mean squared error [40]. The tuning parameters were found via function 'h2o.grid' by running the model with different values for the parameters [40]. They were: maximum

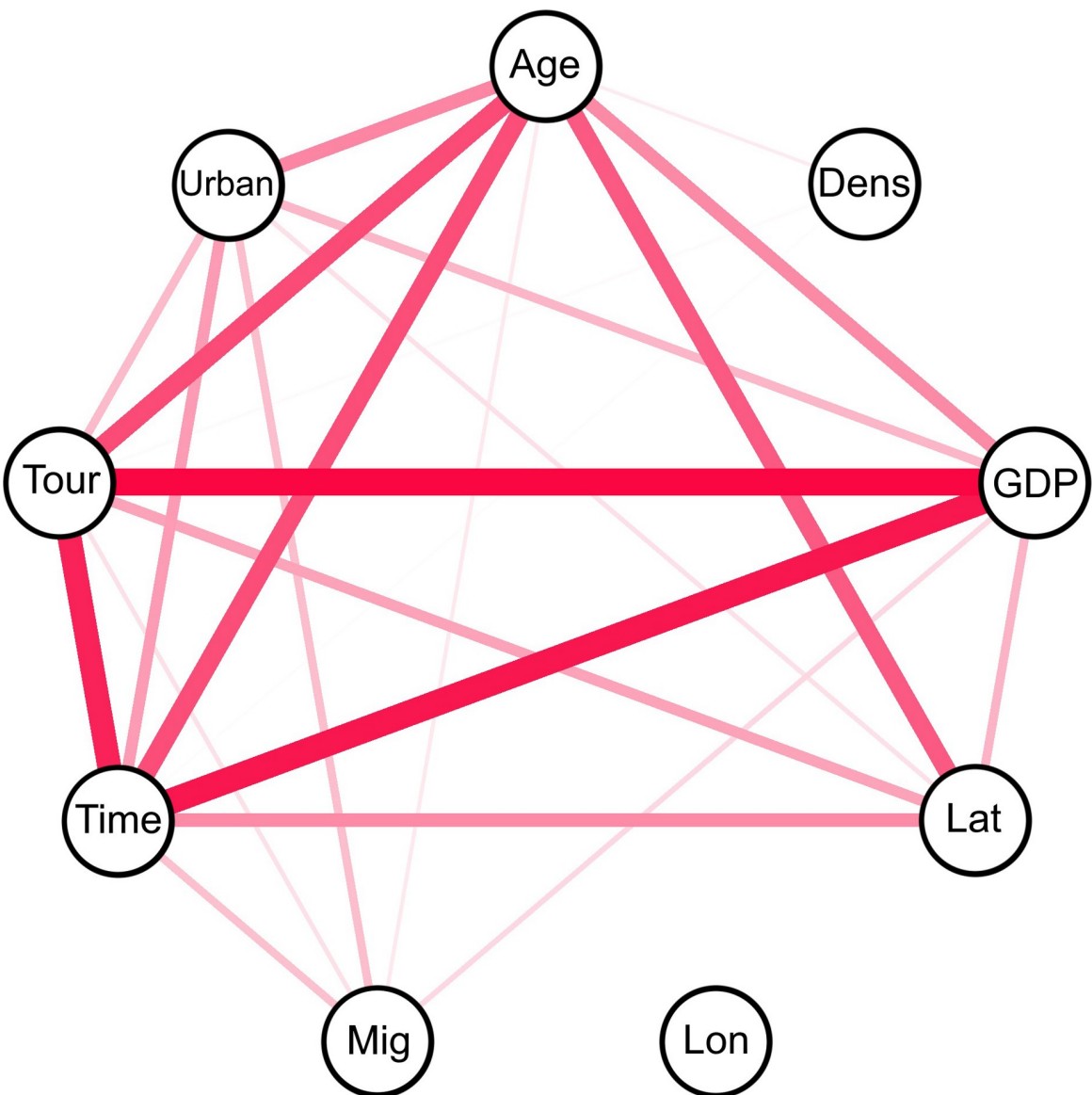

**Fig 1. Correlations among explanatory variables used in the analyses.** Only statistically significant associations are shown. The width of the lines indicates the strength of the correlation. Explanation of variable codes: Age = the median age of the population in a given country; Dens = human population density; GDP = gross domestic product; Lat = geographic latitude of the country centroid; Lon = geographic longitude of the country centroid; Mig = net migration rate; Time = number of days since the start of the pandemic in a given country; Tour = number of tourists in a given country; Urban = the proportion of the human population living in urban areas.

tree depth (values: 1, 3, 5), fewest allowed (weighted) observations in a leaf (values: 1, 5, 10), learning rate (values: 0.001, 0.01, 0.1), scale the learning rate by this factor after each tree (values: 0.99, 1), row sample rate per tree (values: 0.5, 0.75, 1), and column sample rate (values: 0.8, 0.9, 1).

The model was fitted to the training data (70% of data, randomly selected) with 10-fold cross validation [40]. For the number of cases and deaths, we used the Poisson distribution and for the growth rate, we used the Gaussian distribution. We used natural logarithm transformation (variables: Dens, GDP, Time, Tour) because gradient boosting regression may produce biased results in the presence of outliers [44]. The fitted model was then used to make

predictions on the test data. Finally, the performance of each model was assessed on the test dataset. The $R^2$ between the predicted and actual data was used as a measure of performance.

To visualise the results, we used individual conditional expectation (ICE) plots in 'pdp' R package [45], a tool for visualising the model estimated by any supervised learning algorithm and Friedman's partial dependency plots [36]. Partial dependence plot (PDP) highlights the average partial relationship between a set of explanatory variables and the predicted response variable [40]. Individual conditional expectation plots highlight the variation in the fitted values across the range of an explanatory variable, suggesting where and to what extent heterogeneities may exist. The ICE plots disaggregate this average by displaying the estimated functional relationship for each observation [46]. We interpreted the results with an importance above 1%.

## 3. Results

### 3.1. Number of COVID-19 cases

The GBRT analysis revealed that all examined variables had a non-zero impact on the number of cases (Fig 2). However, only three variables, GDP, Tour, and Lon, had an importance above 1% (Fig 2). The number of cases positively correlated with GDP, but in a nonlinear manner (Fig 3A). The number of cases increased after the GDP reached 60 billion US dollars (Fig 3A). The number of cases also increased rapidly if the number of tourists in a country exceeded 20 million (Fig 3B). The number of cases increased with the geographic longitude from Asia to Europe (Fig 3C). Gradient-boosted regression trees built on trained data explained 81% of the variation in the test data.

### 3.2. Number of deaths

The GBRT analysis revealed that all examined variables had a non-zero impact on the number of deaths (Fig 2). However, only four variables, Tour, Cases, GDP, and Lon, had an importance above 1% (Fig 2). The number of deaths increased rapidly if the number of tourists in a country exceeded 30 million (Fig 4A). The number of deaths was positively associated with the number of COVID-19 cases (Fig 4B). The number of deaths increased slightly after the GDP reached 400 billion US dollars (Fig 4C). The number of cases decreased with increasing geographic longitude (Fig 4D). Gradient-boosted regression trees built on trained data explained 92% of the variation in the test data.

### 3.3. Growth in the number of COVID-19 cases

The GBRT analysis revealed that all examined variables had a non-zero impact on the growth rate of COVID-19 cases (Fig 2). The growth rate accelerated with time (Fig 5A). Gross domestic product increased growth rate starting from the values of about 2 billion US dollars, then accelerated if it exceeded 400 billion US dollars (Fig 5B). The growth rate decreased with increasing geographic longitude (Fig 5C). The non-linear effect of population density was found on the growth rate (Fig 5D). The population density with values ranging roughly between 50 and 500 persons per square kilometre decreased the growth rate (Fig 5D). The growth rate of COVID-19 cases decreased with the median age of the country population (Fig 5E). The growth rate changed non-linearly with geographic latitude (Fig 5F). It was elevated between both tropics (Fig 5F). Also, in the northern hemisphere, the countries located above 50˚N had a slower growth rate than countries located more to the south (Fig 5F). The growth rate also increased non-linearly with the number of tourists (Fig 5G). A non-linear effect of the migration rate was found (Fig 5H). Generally, countries with net emigration rates close to zero

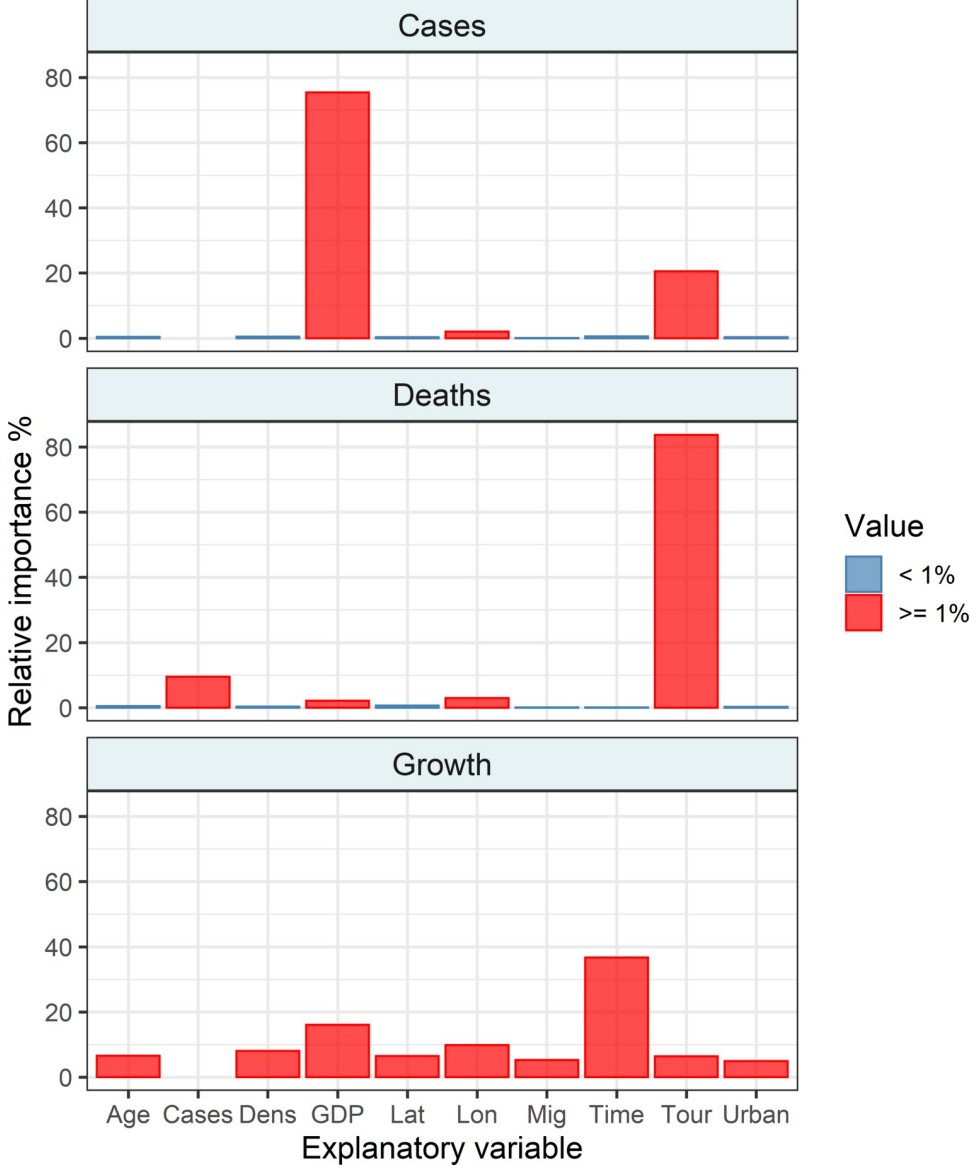

**Fig 2. Decomposition of the variation associated with explanatory variables into independent components using gradient boosting regression trees.** The importance of variables in gradient boosting regression trees explaining the number of COVID-19 cases, number of deaths, and growth rate of COVID-19 cases. Explanatory variables that had the importance of the dependent variables above 1% are given in red. Explanation of variable codes: see Fig 1.

had higher growth rates in the number of COVID-19 cases than countries with both excess immigrants and emigrants (Fig 5G). Finally, the growth rate decreased in countries with a lower proportion of population living in urbanised areas but increased in highly urbanised territories. Gradient-boosted regression trees built on trained data explained 22% of the variation in the test data.

## 4. Discussion

Our macro-ecological approach revealed the impact of several variables shaping the pattern of the COVID-19 pandemic on a global scale. One of our most interesting findings was that we

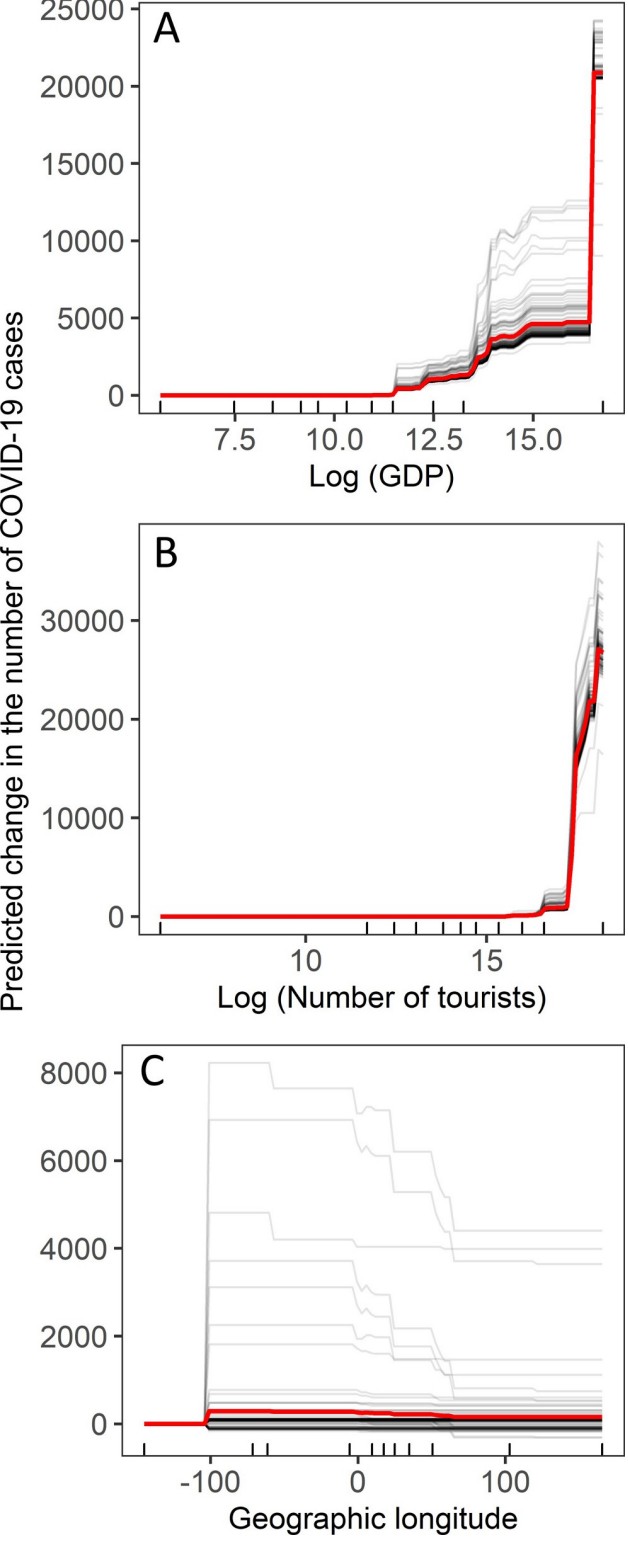

**Fig 3.** Centred individual conditional expectation plots of the predicted number of COVID-19 cases by a) number of tourists, b) gross domestic product, and c) geographic longitude. The lines show the difference in prediction compared with the prediction with the respective value of the explanatory variables at their observed minimum. The red line is the averaged marginal functional estimate from the gradient boosting regression trees. Rug plots inside the bottom of the plots show the distribution of data, in deciles, of the variable on the X-axis.

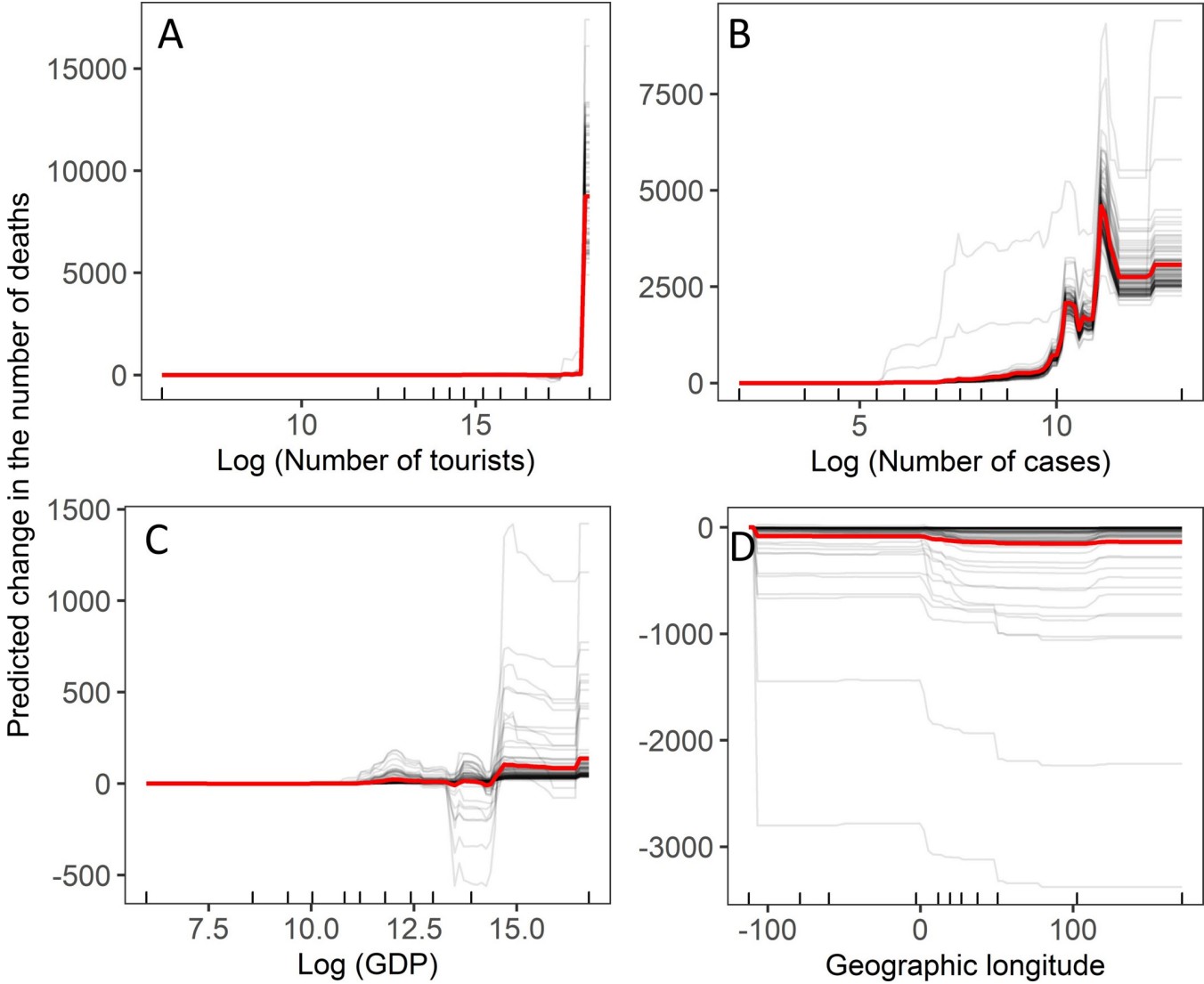

**Fig 4.** Individual conditional expectation plots of the predicted number of deaths by a) number of toursits, b) number of COVID-19 cases, c) gross domestic product, and d) geographic longitude. For explanations, see Fig 3.

did not find evidence of a positive association between population density and infection numbers and deaths. This contradicts our expectations, which were based on theory and earlier findings in other diseases [19,47]. It may be that population density plays a role at lower spatial scales [48]. In addition, human population density in investigated countries is likely to be so high that diseases can easily disperse among people. However, we observed a weak non-linear effect of human population density on growth rate. This effect is also in contradiction to our expectations because the growth rate was low at moderate human densities. This is difficult to explain and possibly other factors not investigated in this study, but linked with population density may obscure this effect.

We found that there is a positive association between the number of tourists visiting a given country and the number of infections, deaths, and growth rate of COVID-19 cases, which is in agreement with our expectations. This indicates that breaking geographical barriers may be a crucial step in colonising new areas and hosts. In ecological terms, the spread of SARS-CoV-2

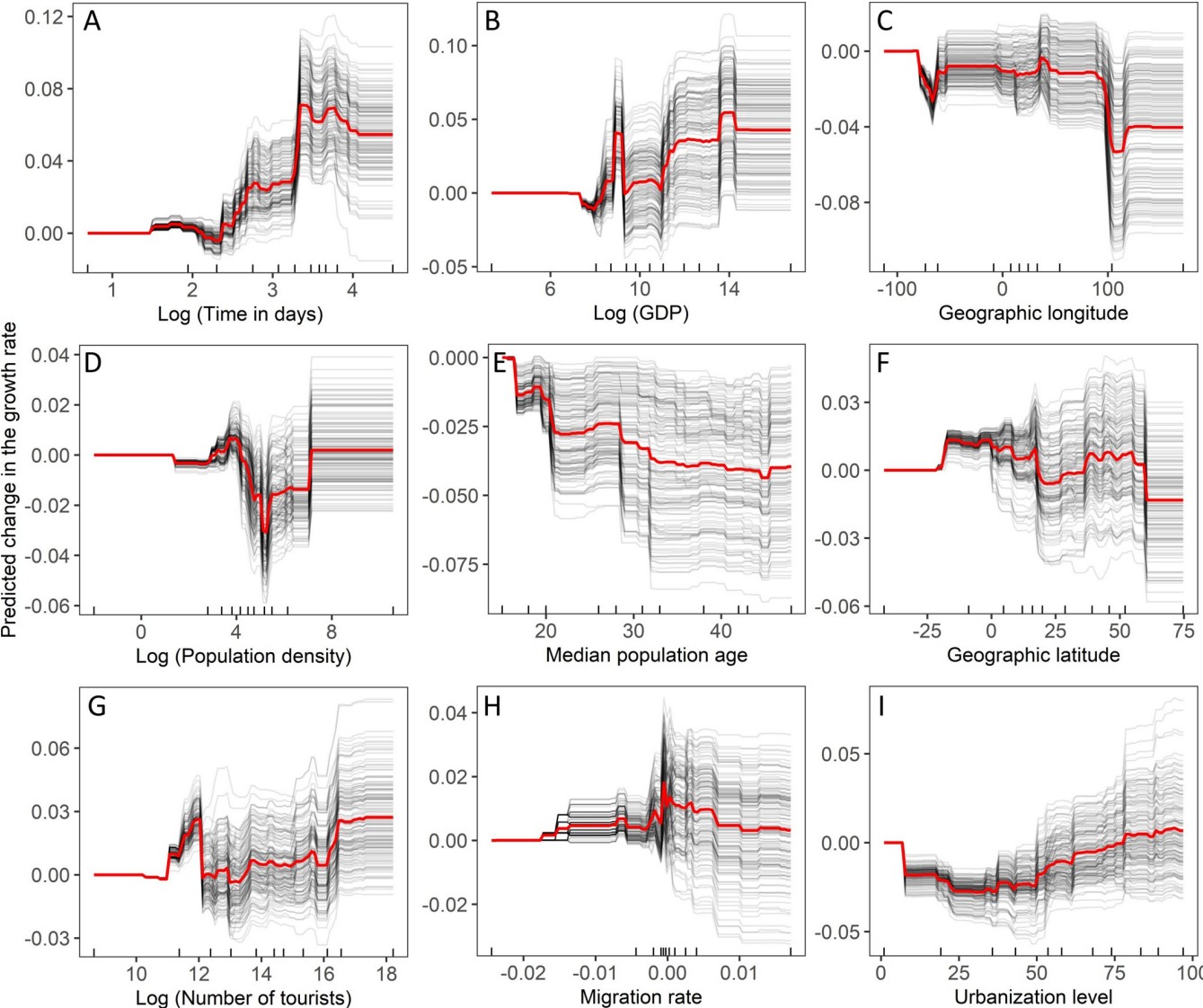

**Fig 5.** Individual conditional expectation plots of the predicted number of deaths by a) number of days since the start of the pandemic in a given country, b) gross domestic product, c) geographic longitude, d) human population density, e) median population age, f) geographic latitude, g) number of tourists, h) migration rate, and i) proportion of human population living in urbanised areas. For further explanations, see Fig 3.

resembles an invasion of an alien species after new geographical areas have been colonised, because of its impact on native ecosystems [49,50]. Overall, the effect was non-linear and the number of tourists had an impact if the number of tourists visiting a given country was high, usually above 20 million. This is also analogous to the invasion process where so-called 'propagule pressure' and continuous colonisations are key triggers of the invasion [50–52]. Global travel has increased in overall number, but there has also been a shift in areas visited by travellers, especially in Asia [53]. The role of tourism in the spread of diseases was reported in previous studies [54]. Early on, the spatial distribution of COVID-19 cases in China was well explained by human mobility data [55]. Thus, it may be that some regulations regarding tourism, such as limited visits to countries with a high prevalence of diseases or quarantine for people returning from them, may indeed be a solution worth considering in this pandemic and

possibly also in future ones. Nevertheless, the role of tourism in the spread of the virus should be investigated thoroughly in future studies because it was one of the most important predictors in our models.

We did not find any impact of the net migration rate on the number of COVID-19 cases and deaths, except for a weak, non-linear association with growth rate of COVID-19 cases. In the latter case, the growth rate was the highest in countries with migration rates close to zero. It is possible that the latter effect involves some biological factors. For example, increased genetic diversity in societies with migrants may be a barrier for pathogens [56], decreasing the chances of virus transmission. However, the net migration rate close to zero may also indicate that immigration and emigration are balanced and this effect may be inseparable from the total isolation. It is important to note that migration substantially differs from touristic trips and is associated with many formal requirements, including health, in some host countries [57,58]. Furthermore, migration is usually a singular event in the life of an individual. Tourism, on the other hand, is linked with much higher mobility, visiting crowded places, and frequent changes of location [57].

Unexpectedly, the gross domestic product was positively related to the number of infections, deaths, and growth rate of the number of virus infections. Worldwide analysis indicated that there is a direct positive relationship between GDP and total health expenditure [59]. There is a positive significant relationship between total health expenditure and increased life expectancy [60]. Moreover, a cohort-based study showed that levels of GDP at the time of death were strongly inversely associated with all-cause mortality, especially among women [61]. However, there is also evidence that higher GDP is linked with morbid behaviours responsible for the occurrence of diseases. Rising income has been strongly associated with higher consumption of unhealthy commodities within countries and over time [62]. In consequence, wealthy, market-liberal countries have more overweight citizens [63] and there is increasing evidence that obesity is an independent risk factor for severe illness and death from COVID-19 [64]. Of course, this relationship may be mutual. Past pandemics, such as the 1918 influenza pandemic, have had a strong negative impact on socioeconomy and gross domestic product [65]. The strong positive association between COVID-19 and gross domestic product indicates that pandemics may strongly affect developed economies, which is in line with the opinions of some experts [66,67].

It is believed that pandemics can be characterized as having low mortality of infected people, high infectivity, a long period of contagiousness, and a lack of natural immunity of the population, and the disease does not destroy its host. Harmless symptoms contribute to neglect of the disease. Coronavirus disease seems to have these characteristics, except for the relatively high mortality among older people [68], mostly due to the prevalence of chronic diseases in older people [69]. However, we did not find an association between median age and number of cases and deaths. We found a relatively weak negative association between growth rate of COVID-19 cases and median age of the population. One possible but risky explanation is that younger people are vectors of the virus, which would be in line with findings for other diseases [35,36]. On the other hand, it was quickly identified that older people are the most endangered group and special care was devoted to older people in the health systems [68]. Thus, actions undertaken by countries could limit the spread of the virus via older people. In addition, older people are usually less mobile with a limited number of social contacts [70], which may explain why viruses may spread slowly in older societies.

We noted the potential impact of urban areas on the growth rate of infection cases. Urban areas are associated with high population density and high levels of social interaction, but also with stress and pollution [20]. This may promote the spread of viruses. Studies on influenza in the United Kingdom in 1918 indicated that death rates varied markedly with urbanisation, with 30% –40% higher rates in cities and towns than in rural areas [71]. However, Wood et al.

[72] found that urbanisation was generally associated with lower burdens for many diseases, a pattern that could arise from increased access to sanitation and healthcare in cities and increased investment in healthcare. Thus, it seems that urban areas may have contradictory effects on transmission according to disease type.

We also found an effect of geographic location on infection rate, mortality, and infection rate. The number of COVID-19 cases and deaths but not growth rate were positively related to geographic longitude. This may be explained by some theoretical studies [73] that found that crossing geographical barriers is a major factor in spreading diseases. However, the decreasing growth rate of the number of cases may reflect the known phenomenon that pandemic spread is the highest in the place of origin and decreases with distance [74].

The growth rate of COVID-19 cases was non-linearly associated with geographic latitude. Geographic position is usually linked to the local climate. Our finding is similar to recent reports [75], with emerging evidence suggesting that weather conditions may influence the transmission of SARS-CoV-2, dry conditions appearing to boost the spread [76]. This phenomenon may manifest itself through two mechanisms: the stability of the virus and the effect of the weather on the host. However, reports indicate that the weather effect is minimal, and all estimates are subject to significant biases, reinforcing the need for robust public health measures [76]. On the other hand, the number of contacts among people may also be affected by climate. People born in a warmer climate are much more social than those coming from cold regions [77]. This may create opposing forces on the spread of the virus. We believe that further models that include more precise geolocation of infection data and local climatic and local human population density are highly warranted.

Not surprisingly, the growth rate of COVID-19 cases was positively associated with time. This variable is usually the most important factor in predicting the number of infections and diseases [78,79]. However, this variable is especially important if there is a time lag between incidence and healthcare system response with possible consequences for virus spread dynamics in space [80,81].

## 4.1. Study limitations

Our study has certain limitations that must be taken into consideration. Our analysis is based on data from the early stages of the pandemic. Repeated analyses after several weeks may yield different results. For example, different variables may play a role in different pandemic stages [82]. Moreover, our study is, of course, correlative. Thus, associations between explanatory variables and dependent variables should be treated with caution. Moreover, our analyses are based on 'big data', which is known to have caveats [83]. For example, the positive association between GDP and the number of COVID-19 cases may result from better diagnostics and a large number of performed tests in rich countries. Moreover, GDP is associated with many other variables and real-world phenomena [3]. Thus, this association should be interpreted with caution. Finally, our explanatory variables were correlated with each other. However, the values of correlation were moderate and the GBRTs were more robust in multicollinearity situations than ordinary least squares regression and produce reliable estimates that were straightforward to interpret in partial dependency plots. Nevertheless, only experimental tests of our hypotheses on non-human organisms would result in cause-effect relationships. However, studies on a global scale rarely, if at all, are experiments.

## 4.2. Conclusions

The COVID-19 pandemic prompted the need to identify the important components in the disease spread for better projections of global-scale pandemics. Several factors, such as

anthropogenic environmental changes, human demography, international travel, and microbial adaptation, probably have contributed to the disease with which the global community is currently challenged. Unfortunately, epidemics seem to be idiosyncratic, which makes prediction much harder. However, if pathogen spread is a result of understood intrinsic processes, the relationships can be incorporated into pandemic predictions and healthcare response and delivery. This would require political agreement and cooperation in the exchange of information and open access to all data on diseases. Moreover, a multidisciplinary and macroscale approach [2] is needed, both in research and policymaking to better control and monitor the spread of diseases. Last but not least, the Anthropocene was proposed to delineate the epoch of significant human impact on Earth's ecosystems (e.g. climate change) [4,5,84]. The COVID-19 pandemic shows that the impact may be altered by a virus and raises the question of whether human impact is longstanding. Nevertheless, the positive correlation between infection number, deaths, and gross domestic product suggests that COVID-19 may be a new civilisation disease.

## Supporting information

**S1 File. Covid_19–contains all the data used in analyses.**
(XLSX)

**S2 File. Covid_19_codes–contains codes to reproduce the results.** Codes used data from the file Covid_19.
(R)

## Acknowledgments

We thank the anonymous referee for the constructive comments on earlier versions of this manuscript.

## Author Contributions

**Conceptualization:** Piotr Skórka, Beata Grzywacz, Dawid Moroń, Magdalena Lenda.

**Data curation:** Piotr Skórka, Beata Grzywacz.

**Formal analysis:** Piotr Skórka, Dawid Moroń.

**Methodology:** Piotr Skórka, Beata Grzywacz, Dawid Moroń, Magdalena Lenda.

**Resources:** Magdalena Lenda.

**Software:** Piotr Skórka.

**Supervision:** Beata Grzywacz, Magdalena Lenda.

**Validation:** Beata Grzywacz, Dawid Moroń, Magdalena Lenda.

**Visualization:** Piotr Skórka.

**Writing – original draft:** Piotr Skórka, Beata Grzywacz, Dawid Moroń, Magdalena Lenda.

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
