## [Decision Letter · Decision Letter 0]

22 May 2020

PONE-D-20-08769

The macro-ecology of COVID-19 pandemics in Anthropocene

PLOS ONE

Dear Dr. Skórka,

Thank you for submitting your manuscript to PLOS ONE. After careful consideration, we feel that it has merit but does not fully meet PLOS ONE’s publication criteria as it currently stands. Therefore, we invite you to submit a revised version of the manuscript that addresses the points raised during the review process.

We look forward to receiving your revised manuscript.

Kind regards,

Abdallah M. Samy, PhD

Academic Editor

PLOS ONE

**Journal Requirements:**

**2. **We note that you have reported significance probabilities of 0 in places. Since p=0 is not strictly possible, please correct this to a more appropriate limit, eg 'p<0.0001'.

**Reviewers' comments:**

Reviewer's Responses to Questions

**Comments to the Author**

1. Is the manuscript technically sound, and do the data support the conclusions?

Reviewer #1: Partly

2. Has the statistical analysis been performed appropriately and rigorously? 

Reviewer #1: Yes

3. Have the authors made all data underlying the findings in their manuscript fully available?

Reviewer #1: Yes

4. Is the manuscript presented in an intelligible fashion and written in standard English?

Reviewer #1: No

5. Review Comments to the Author

Reviewer #1: This paper considers the factors that are correlated with country level variation in the number of cases, number of deaths and growth rate of COVID-19 infections. It is a timely paper on an interesting topic that synthesizes data from a variety of sources. The data that the authors have gotten together alone will be of value to researchers. However, I find that in the current form it contains a somewhat confusing admixture of results. There are also some minor issues with the prose.

Major Issue

The primary issue I struggled with was interpreting the statistical results. The results from univariate models, multivariate models and hierarchical partitioning differed greatly. The authors place the most emphasis on the latter results (e.g., the four variables that come up in both the cases and deaths panel of figure three are the ones that are mentioned in the abstract). However, I think the majority of readers will be most familiar with the methods shown in Table 1 and Table 2. I found the differences between these three sets of results confusing, and I did not find that the authors made a good case for why the results from Fig. 3 are more informative than the multivariate models (save the obvious fact that more variables show statistically significant effects). For example, age and number of tourists showed no significant effects in Table 2, but show up as relatively important in Figure 3. GDP is probably is the single most important predictor to include in models of number of deaths based on hierarchical partitioning (Fig. 3), but isn’t anywhere near being significant in multivariate models of number of deaths (Table 2). The fact that the relative importance of predictors changes so much from Table 1, to Table 2 to Figure 3 is also disconcerting. It implies that the results the authors discuss are not very robust.

At the very least, the authors will need to make a clear case for why hierarchical partitioning is the most useful and informative method for these analyses. I also wonder if some other modelling framework might be more informative. For example, gams can be implemented using the mgcv package (wood and wood, 2015), and boosted regression trees can easily be implemented using gbm (Ridgeway 2013). Both methods would allow for the discovery of nonlinear relationship between response and predictor variables (which also relates to a more minor issue, below), and the latter method is also robust to the use of collinear predictors with complex interaction effects. In fact the way the gbm calculates relative influence scores is very similar to the logic of hierarchical partitioning, and I believe more readers would be familiar with the method.

Minor Issues

1. It does not appear that the methods the authors use would allow them to detect nonlinear effects. For example, they found no influence of human population density on number or rate of infections. If it exhibits a threshold effect rather than a linear effect, it might be difficult to detect with methods that assume a linear relationship. For example, the authors speculate that the countries included had high enough population densities that COVID-19 can always easily spread in them (Page 6, lines 215-216). To me this implies that most of the countries included are above some key threshold. If there are at least a few countries below the threshold, a marginal plot from a gbm model or a gam plot would show this pattern clearly.

The fact that GDP is overall positively correlated with number of deaths and infections is also puzzling. I found the authors' discussion of this result on page seven interesting. However, I wonder if this might be a case where a nonlinear relationship occurs that might be easier to explain (for example, international trade and travel and thus risk increases up to some threshold of GDP, but then further increases in per capita GDP actually do slightly lower death rates).

2. I am not sure most readers will know what the authors mean by macroecology. It is used in the title but never defined. I assume the authors mean macroecology sensu Brown et al. (2014) and Burnsdie et al. (2012), but I’m not entirely sure. If this is what the authors intend, these or some other studies clarifying the relevance of the term to their work should at least be mentioned.

3. There are also a lot of minor issues with the writing (mainly typos). To illustrate this I am going to give examples, from the first few pages, but this list is nowhere near comprehensive.

Page 1, line 12: “Covid-19 has expanded” or “The COVID-19 virus has expanded” would be ok. “The COVID-19 has expanded” doesn’t quite make sense.

Page 2, line 43: Should be “those caused by”

Page 2 line 46: missing “economy” should be plural (“economies”)

Page 2 lines 47-50: Need a citation

Page 2 line 52: “the incidence occurred mainly in the city of Wuhan” is a bit unclear. Maybe the authors are trying to say “cases occurred mainly in the city of Wuhan” or “incidence was limited primarily to the city of Wuhan”

Page 2 line 54: “the first case in another country outside China” If the cases were in another country, of course they were outside China. I think “the first case outside of China was confirmed” would be much clearer.

Page 3 line 84” Models predict that most children are responsible for transmission of the virus.” I am fairly certain that most children have yet to encounter the virus, much less transmit it. Are the authors trying to say that the virus is primarily transmitted by children? Even if this is what the authors intended to convey, I am not really sure that’s true given that the great majority of confirmed cases are in adults.

Page 3, lines 106-122: Every single one of these starts with “The number of infections, deaths, and the rate of growth in the number of COVID-19 infections are”. This section would be much easier to read if that phrase only occurred once at the beginning of the section (around line 107). For example.

“We tested the hypotheses that the number of infections, deaths, and the rate of growth in the number of COVID-19 infections are:

1. Positively associated with human population density.

2. Positively associated with the proportion of the population living in urban areas.”

3. ect."

Cited:

Brown, James H., Joseph R. Burger, William R. Burnside, Michael Chang, Ana D. Davidson, Trevor S. Fristoe, Marcus J. Hamilton et al. "Macroecology meets macroeconomics: Resource scarcity and global sustainability." Ecological engineering 65 (2014): 24-32.

Burnside, William R., James H. Brown, Oskar Burger, Marcus J. Hamilton, Melanie Moses, and Luis MA Bettencourt. "Human macroecology: Linking pattern and process in big‐picture human ecology." Biological Reviews 87, no. 1 (2012): 194-208.

Ridgeway, Greg, Maintainer Harry Southworth, and Suggests RUnit. "Package ‘gbm’." Viitattu 10, no. 2013 (2013): 40.

Wood, S., & Wood, M. S. (2015). Package ‘mgcv’. R package version, 1, 29.

6. PLOS authors have the option to publish the peer review history of their article (what does this mean?). If published, this will include your full peer review and any attached files.

Reviewer #1: No

---

## [Author Response · Author response to Decision Letter 0]

2 Jul 2020

Dear Editor,

I would like to submit our revised manuscript PONE-D-20-08769R1 “The macro-ecology of the COVID-19 pandemic in the Anthropocene” by Piotr Skórka, Beata Grzywacz, Dawid Moroń, Magdalena Lenda.

We are grateful for all critical points that helped us to improve our paper. We did our best to incorporate all critical points in the revised version. First, we changed statistical analysis into gradient boosting regression as was suggested by the reviewer. We also added newer data that enabled us to increase sample size and receive a better picture of the pandemics. A new analysis produced less results that are easier for interpretation. We however had to change discussion as fewer variables appeared to be meaningful in analyses. We also better described the theoretical background of our paper by defining explicitly the term “macro-ecology”. We also corrected all minor issues. Moreover, the entire text was linguistically corrected by a native English-man familiar with scientific writing.

 We believe that our revised manuscript meets scientific criteria required for publication in PloS One. 

With kind regards

on behalf of the authors,

Piotr Skórka

5. Review Comments to the Author

Reviewer #1: This paper considers the factors that are correlated with country level variation in the number of cases, number of deaths and growth rate of COVID-19 infections. It is a timely paper on an interesting topic that synthesizes data from a variety of sources. The data that the authors have gotten together alone will be of value to researchers. However, I find that in the current form it contains a somewhat confusing admixture of results. There are also some minor issues with the prose.

RESPONSE: Thank you for your assessment of our work. We did our best to incorporate all points and suggestions into the revised manuscript.

Major Issue

The primary issue I struggled with was interpreting the statistical results. The results from univariate models, multivariate models and hierarchical partitioning differed greatly. The authors place the most emphasis on the latter results (e.g., the four variables that come up in both the cases and deaths panel of figure three are the ones that are mentioned in the abstract). However, I think the majority of readers will be most familiar with the methods shown in Table 1 and Table 2. I found the differences between these three sets of results confusing, and I did not find that the authors made a good case for why the results from Fig. 3 are more informative than the multivariate models (save the obvious fact that more variables show statistically significant effects). For example, age and number of tourists showed no significant effects in Table 2, but show up as relatively important in Figure 3. GDP is probably is the single most important predictor to include in models of number of deaths based on hierarchical partitioning (Fig. 3), but isn’t anywhere near being significant in multivariate models of number of deaths (Table 2). The fact that the relative importance of predictors changes so much from Table 1, to Table 2 to Figure 3 is also disconcerting. It implies that the results the authors discuss are not very robust.

At the very least, the authors will need to make a clear case for why hierarchical partitioning is the most useful and informative method for these analyses. I also wonder if some other modelling framework might be more informative. For example, gams can be implemented using the mgcv package (wood and wood, 2015), and boosted regression trees can easily be implemented using gbm (Ridgeway 2013). Both methods would allow for the discovery of nonlinear relationship between response and predictor variables (which also relates to a more minor issue, below), and the latter method is also robust to the use of collinear predictors with complex interaction effects. In fact the way the gbm calculates relative influence scores is very similar to the logic of hierarchical partitioning, and I believe more readers would be familiar with the method.

RESPONSE: We are grateful for these comments and suggestions. Indeed, results of these three analyses differed greatly. We believe this is a result of positive correlation among variables and it is known that multiple regression may produce biased estimates (despite variance inflation factors were acceptable). We really like the idea of using gradient boosting regression (gbm) proposed by the referee. We had not used this method before. Indeed, this method copes with collinearity among predictors by producing importance scores and partial dependency plots for each variable. Also, as it was mentioned by the Reviewer the method allows to identify nonlinear relationships among variables. Therefore, we decided to use the gradient boosting machine learning technique to analyse results. First of all, we decided to add more newer data (gathered on 12th April) on COVID-19. This was done to enlarge sample size (larger number of countries of data) and get better estimates of the pandemic growth rates (but still with exponential mode). We used h2o.gbm function from h2o package (LeDell et al. 2020) because it enabled better visualization of results than ‘gbm` package by easier production of ice plots (individual conditional expectation plots). We searched for optimal parameters to build regression trees in this method. The use of advantage of gradient boosting regression was that it produced one set of results for each dependent variable. Moreover, it omits problems with P-values which use is being criticized very often. Results were slightly different, we identified a lower number of important variables, however the most interesting results, e.g. positive effect of growth domestic product and number of tourists in a country on the COVID-19 spread, remained. 

Erin LeDell, Navdeep Gill, Spencer Aiello, Anqi Fu, Arno Candel, Cliff Click, Tom Kraljevic, Tomas Nykodym, Patrick Aboyoun, Michal Kurka and Michal Malohlava (2020). h2o: R Interface for the 'H2O' Scalable Machine Learning Platform. R package version 3.30.0.1. https://CRAN.R-project.org/package=h2o

Minor Issues

1. It does not appear that the methods the authors use would allow them to detect nonlinear effects. For example, they found no influence of human population density on number or rate of infections. If it exhibits a threshold effect rather than a linear effect, it might be difficult to detect with methods that assume a linear relationship. For example, the authors speculate that the countries included had high enough population densities that COVID-19 can always easily spread in them (Page 6, lines 215-216). To me this implies that most of the countries included are above some key threshold. If there are at least a few countries below the threshold, a marginal plot from a gbm model or a gam plot would show this pattern clearly.

RESPONSE: We agree. However, in ‘gbm’ models the effect of population density was identified as unimportant (which was somehow a surprise to us).

The fact that GDP is overall positively correlated with number of deaths and infections is also puzzling. I found the authors' discussion of this result on page seven interesting. However, I wonder if this might be a case where a nonlinear relationship occurs that might be easier to explain (for example, international trade and travel and thus risk increases up to some threshold of GDP, but then further increases in per capita GDP actually do slightly lower death rates).

RESPONSE: We agree. The new analysis revealed exactly what was said by the Reviewer. We included these explanations also in the revised version of our manuscript.

2. I am not sure most readers will know what the authors mean by macroecology. It is used in the title but never defined. I assume the authors mean macroecology sensu Brown et al. (2014) and Burnsdie et al. (2012), but I’m not entirely sure. If this is what the authors intend, these or some other studies clarifying the relevance of the term to their work should at least be mentioned.

RESPONSE: Thank you for these interesting works. We wrote a paragraph about human macroecology in Introduction and cited the abovementioned publications.

3. There are also a lot of minor issues with the writing (mainly typos). To illustrate this I am going to give examples, from the first few pages, but this list is nowhere near comprehensive.

RESPONSE: We apologize for these mistakes. The revised version of the manuscript was linguistically corrected by native English-man from Wiley Authors Service. We hope the revised version if free of such problems.

Page 1, line 12: “Covid-19 has expanded” or “The COVID-19 virus has expanded” would be ok. “The COVID-19 has expanded” doesn’t quite make sense.

RESPONSE: We changed this sentence to be more specific: “The SARS-CoV-2 coronavirus, causing coronavirus disease 2019 (COVID-19), has expanded…”

Page 2, line 43: Should be “those caused by”

RESPONSE: Corrected.

Page 2 line 46: missing “economy” should be plural (“economies”)

RESPONSE: Corrected.

Page 2 lines 47-50: Need a citation

RESPONSE: Citation added. 

Page 2 line 52: “the incidence occurred mainly in the city of Wuhan” is a bit unclear. Maybe the authors are trying to say “cases occurred mainly in the city of Wuhan” or “incidence was limited primarily to the city of Wuhan”

RESPONSE: We clarified this sentence.

Page 2 line 54: “the first case in another country outside China” If the cases were in another country, of course they were outside China. I think “the first case outside of China was confirmed” would be much clearer.

RESPONSE: Changed as suggested.

Page 3 line 84” Models predict that most children are responsible for transmission of the virus.” I am fairly certain that most children have yet to encounter the virus, much less transmit it. Are the authors trying to say that the virus is primarily transmitted by children? Even if this is what the authors intended to convey, I am not really sure that’s true given that the great majority of confirmed cases are in adults.

RESPONSE: We were not clear in this sentence. We meant that generally viruses (not COVID-19) are transmitted by children (there is a good body of literature on this). We corrected these sentences. 

Page 3, lines 106-122: Every single one of these starts with “The number of infections, deaths, and the rate of growth in the number of COVID-19 infections are”. This section would be much easier to read if that phrase only occurred once at the beginning of the section (around line 107). For example.

“We tested the hypotheses that the number of infections, deaths, and the rate of growth in the number of COVID-19 infections are:

1. Positively associated with human population density.

2. Positively associated with the proportion of the population living in urban areas.”

3. ect."

RESPONSE: We agree and corrected as suggested. Thank you.

Cited:

Brown, James H., Joseph R. Burger, William R. Burnside, Michael Chang, Ana D. Davidson, Trevor S. Fristoe, Marcus J. Hamilton et al. "Macroecology meets macroeconomics: Resource scarcity and global sustainability." Ecological Engineering 65 (2014): 24-32.

Burnside, William R., James H. Brown, Oskar Burger, Marcus J. Hamilton, Melanie Moses, and Luis MA Bettencourt. "Human macroecology: Linking pattern and process in big‐picture human ecology." Biological Reviews 87, no. 1 (2012): 194-208.

Ridgeway, Greg, Maintainer Harry Southworth, and Suggests RUnit. "Package ‘gbm’." Viitattu 10, no. 2013 (2013): 40.

Wood, S., & Wood, M. S. (2015). Package ‘mgcv’. R package version, 1, 29.

Thank you.

---

## [Editor Report · Decision Letter 1]

16 Jul 2020

The macroecology of the COVID-19 pandemic in the Anthropocene

PONE-D-20-08769R1

Dear Dr. Skórka,

We’re pleased to inform you that your manuscript has been judged scientifically suitable for publication and will be formally accepted for publication once it meets all outstanding technical requirements.

Kind regards,

Abdallah M. Samy, PhD

Academic Editor

PLOS ONE

---

## [Editor Report · Acceptance letter]

22 Jul 2020

PONE-D-20-08769R1 

The macroecology of the COVID-19 pandemic in the Anthropocene 

Dear Dr. Skórka:

I'm pleased to inform you that your manuscript has been deemed suitable for publication in PLOS ONE. Congratulations! Your manuscript is now with our production department. 

Kind regards, 

on behalf of

Dr. Abdallah M. Samy 

Academic Editor

PLOS ONE